# Filtering-Based Three-Axis Attitude Determination Package for Spinning Spacecraft: Preliminary Results with Arase

**Halil Ersin Soken** [1,*] , **Shin-ichiro Sakai** [2], **Kazushi Asamura** [2], **Yosuke Nakamura** [2], **Takeshi Takashima** [2] **and Iku Shinohara** [2]

[1]   Tubitak Space Technologies Research Institute, Ankara 06800, Turkey
[2]   Japan Aerospace Exploration Agency, Sagamihara 252-5210, Japan; sakai@isas.jaxa.jp (S.S.);
      asamura@stp.isas.jaxa.jp (K.A.); nakamura.yosuke@jaxa.jp (Y.N.); ttakeshi@stp.isas.jaxa.jp (T.T.);
      iku@stp.isas.jaxa.jp (I.S.)
*   Correspondence: ersin.soken@tubitak.gov.tr

**Abstract:** JAXA's ERG (Exploration of Energization and Radiation in Geospace) Spacecraft, which is nicknamed Arase, was launched on 20 December 2016. Arase is a spin-stabilized and Sun-oriented spacecraft. Its mission is to explore how relativistic electrons in the radiation belts are generated during space storms. Two different on-ground attitude determination algorithms are designed for the mission: A TRIAD-based algorithm that inherits from old missions and a filtering-based new algorithm. This paper, first, discusses the design of the filtering-based attitude determination algorithm, which is mainly based on an Unscented Kalman Filter (UKF), specifically designed for spinning spacecraft (SpinUKF). The SpinUKF uses a newly introduced set of attitude parameters (i.e., spin-parameters) to represent the three-axis attitude of the spacecraft and runs UKF for attitude estimation. The paper then presents the preliminary attitude estimation results for the spacecraft that are obtained after the launch. The results are presented along with the encountered challenges and suggested solutions for them. These preliminary attitude estimation results show that the expected accuracy of the fine attitude estimation for the spacecraft is less than 0.5°.

**Keywords:** attitude determination; spinning spacecraft; Arase; attitude filter

## 1. Introduction

The JAXA's Arase (called ERG—Exploration of Energization and Radiation in Geospace—before the launch) spacecraft was launched on 20 December 2016. It is a small satellite mission into geospace which especially focuses on the relativistic electron acceleration, as well as the dynamics of the space storms [1,2]. High-energy particles (ions and electrons) are trapped in the Earth's magnetic field and formed the Van Allen radiation belts. The mission's aim is to elucidate acceleration and loss mechanisms of these highly-charged particles in the inner magnetosphere during space storms.

Arase is Sun-oriented and spin-stabilized about the major-axis with ~7.5 rpm spin rate (spin period of 8 s). The spin-axis, which is defined as the body Z axis, is required to be 5° ~ 15° off of the Sun direction. The spacecraft weighs about 350 kg. It has four solar array panels, two 5 m long masts and four 15 m long wire antennas, which altogether introduce flexibility to the overall spacecraft structure (Figure 1). The spacecraft is located on a highly elliptical orbit with a perigee about 440 km and an apogee about 32,000 km. One orbital period lasts about 9.5 h. The orbit inclination is 32°, which is almost same as the latitude of the launch site [2].

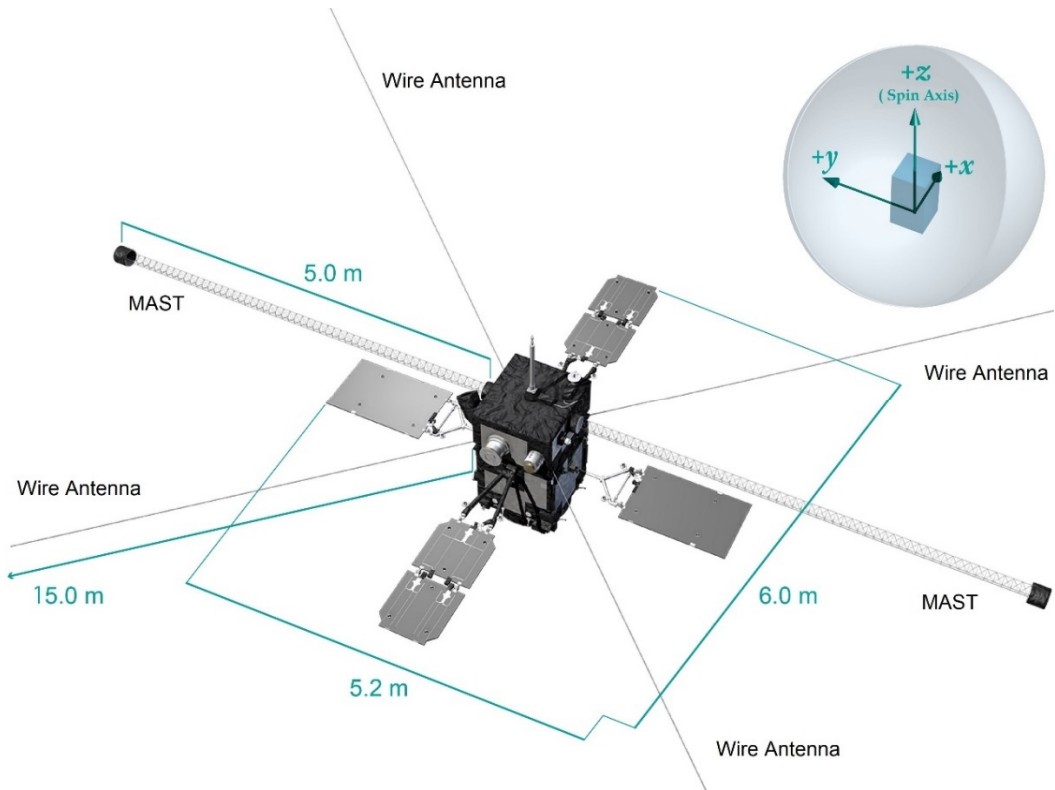

**Figure 1.** Arase and its configuration in the nominal mission phase (courtesy of ISAS/JAXA).

The attitude determination functions are performed on-ground. Arase has three types of attitude sensors onboard to estimate the spin-axis direction (Figure 2), a three-axis fluxgate magnetometer (termed as Geomagnetic Aspect Sensor—GAS), two spin-type Sun aspect sensors (SSAS) and a star scanner (SSC). Only one of the SSASs is operational during the nominal mission. There are also MEMS gyros onboard the spacecraft but they are only used for the initial spin rate and nutation control after the separation from the launcher.

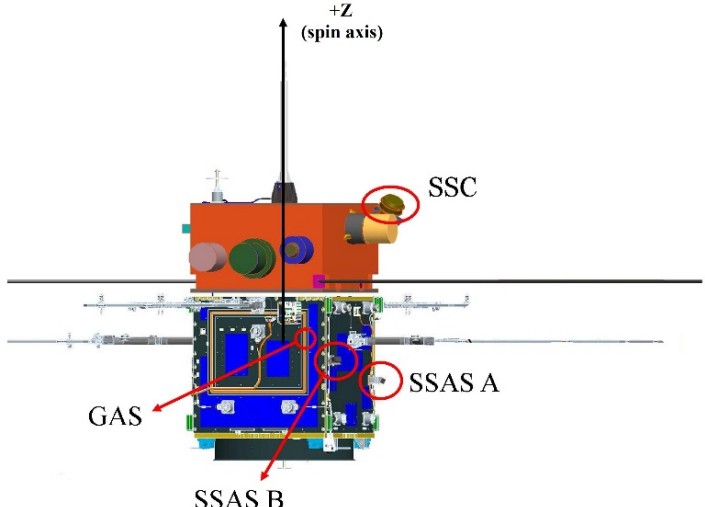

**Figure 2.** Attitude sensors.

During the nominal mission the attitude is controlled using the magnetic torquers (MTQs) to correct deviation of the spin-axis from the intended direction. Attitude maneuvers are performed

by uploading the desired magnetic moments. The reaction control system (RCS), built in four monopropellant hydrazine thrusters, are used for attitude maneuvers in the initial mission phase.

Two on-ground attitude determination algorithms are used for the mission: A TRIAD-based algorithm that inherits from old JAXA missions and a newly designed filtering-based algorithm. The TRIAD-based attitude determination algorithm applies TRIAD [3] and a weighted-averaging method, similar to QUEST [4], in order to estimate the attitude whenever the sensor measurements are available. On the other hand, the filtering-based attitude determination algorithm estimates the spacecraft's attitude using the SpinUKF [5]. SpinUKF is an Unscented Kalman Filter (UKF) algorithm for which the spacecraft's attitude is represented with the so called spin parameters. This new attitude determination package is also capable of calibrating the on-board magnetometers and SSAS and estimating the spin-axis tilt for the spacecraft.

The design of the filtering-based attitude determination package was presented previously in [6,7]. In this paper, we review the algorithm and give the preliminary attitude estimation results for the spacecraft that we obtained after the launch. This includes results for magnetometer and SSAS calibration, which are used to improve the attitude knowledge. The results for each sub-algorithm are presented along with the challenges we had and the suggested solutions. Specifically, we evaluate the performance of the newly proposed algorithms such as the SpinUKF. The initial results, which were presented in [7] in part, are updated. We also provide more comprehensive discussion on the encountered challenges and the suggested solutions.

In the next section, we present an overview for the filtering-based attitude determination package. Then in the third section, we give the magnetometer calibration algorithms and the calibration results. In the fourth section, we provide the coarse attitude estimation results. In the fifth section, we briefly present the star identification algorithm. In the sixth section, we present the fine attitude estimation results for the spacecraft.

## 2. The Filtering-Based Attitude Determination Package

The proposed filtering-based attitude determination package for Arase is composed of three main parts: The coarse attitude estimator, star identification and tracking algorithm, and the fine attitude estimator (Figure 3). The algorithms are executed in the order we listed them: First, the coarse attitude is estimated, then the star identification is performed, and finally, using the identified SSC measurements, the fine attitude is estimated.

Similar to the Multimission Spin Axis Stabilized Spacecraft (MSASS) software package provided by the NASA Goddard Space Flight Center [8,9] the proposed filtering-based attitude determination package is capable of estimating three-axis attitude of a spin-stabilized spacecraft and performing multiple other tasks including the spin-axis tilt estimation. Among the algorithms, the real-time magnetometer calibration algorithm, the SpinUKF for attitude estimation and the spin-axis tilt estimation algorithm are novel algorithms.

The coarse attitude is estimated using the GAS and SSAS measurements. TRIAD algorithm is mainly used to estimate the coarse spin-axis direction. The coarse spin-axis estimate is one of the inputs for the star identification at the later stage together with the calculated spin rate.

The sampling period for the collected GAS measurements is 0.125 s. Although, it is possible to collect the GAS measurements with a higher sampling frequency, 0.125 s is selected considering the data size. The time required for processing the measurements, which must be repeated every day, is another criterion for this selection. The SSAS provides Sun aspect angle (angle between the spin-axis and the Sun direction) at every Sun crossing, which is once every 8 s (at the spin rate of 7.5 rpm).

The GAS measurements are calibrated using two different algorithms. The TWO-STEP algorithm [10] is used for batch calibration. In the end, using the whole available GAS measurements, it provides single estimate for the vector of full calibration parameters: scale factors, nonorthogonality corrections and biases. The other magnetometer calibration algorithm, a pseudo-linear Kalman filter (PSLKF) [11], estimates the time-varying bias components, specifically over the perigee passes. It is a

new algorithm, first proposed in [11]. In this article, we give the results for the algorithm with real data from Arase.

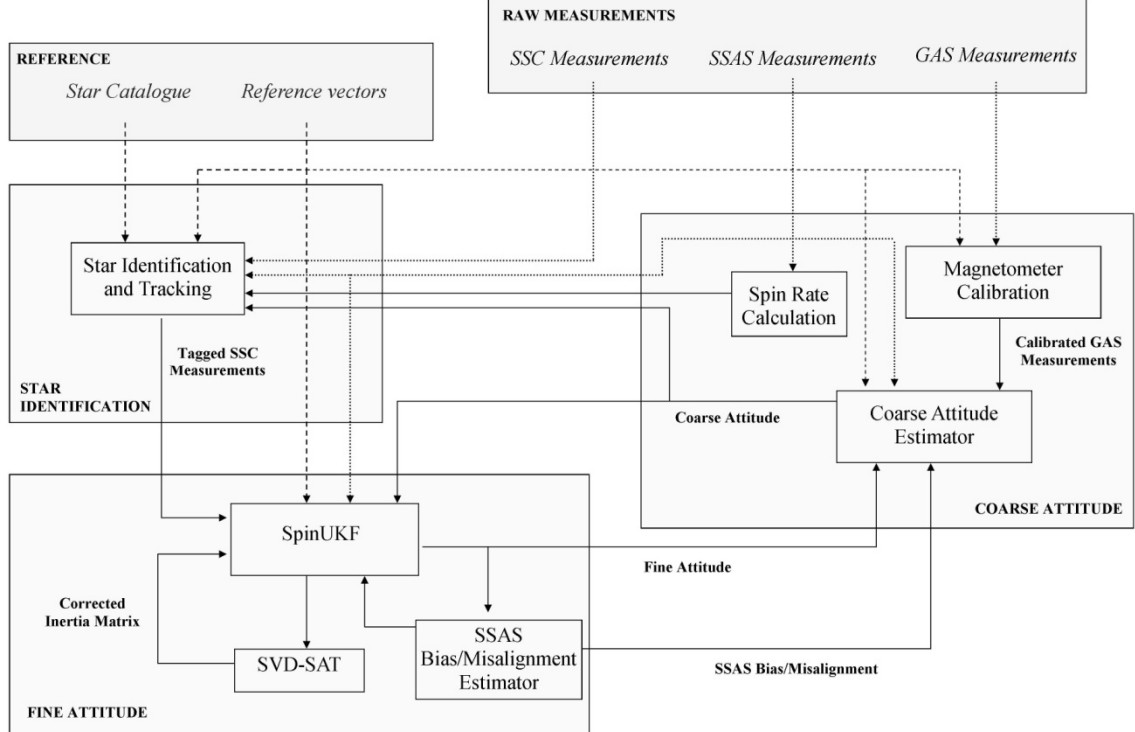

**Figure 3.** Overview of the filtering-based attitude determination package.

Fine attitude estimation for Arase relies on the SSAS and SSC measurements. The SSC onboard the Arase cannot autonomously identify the detected stars. Instead, it provides the star crossing times for each of its slits in a V-slit configuration. Considering the configuration details for the sensor slits, the star aspect angle and a single star crossing time, which can then be used to obtain the star direction vector in the sensor frame, are calculated. In addition, the SSC gives the magnitude of the detected pulses, which indicates the apparent magnitude for the detected stars.

The star identification algorithm first pairs the star pulses detected by the SSC to eliminate the possible misdetections. Then, the algorithm uses the coarse attitude estimate to match the detected pulses with the catalogue stars. The identification process applies the distance-orientation algorithm for star matching [12]. The overall procedure, used for star identification, which includes pairing, identification and tagging stages, is unique and specifically designed for identifying the star scanner vector measurements that is provided whenever a star is detected.

The fine attitude estimator consists of three algorithms. First one of these is UKF for spinning spacecraft (SpinUKF). Previous examples of UKF algorithm for spinning spacecraft attitude estimation are inspired by their counterparts for three-axis stabilized spacecraft [13]. Only in [14] the UKF is implemented with angular-momentum-based state vector, which is derived for spinning spacecraft. SpinUKF is tested with real data in this article is a new algorithm, as well as specifically designed for spinning spacecraft attitude estimation. It uses spin parameters in its state vector, and estimates the full three-axis attitude of the spacecraft along with the attitude rates, sequentially [5]. Spin parameters consist of the spin-axis orientation unit vector in the inertial frame and the spin phase angle (angle about the body spin-axis). This representation is advantageous as the spin-axis direction components in the inertial frame do not change rapidly, and the phase angle changes with a constant rate in the absence of a torque.

The second algorithm for fine attitude estimation is the SAT estimation algorithm [15]. It estimates the SAT error, which is also known as dynamic imbalance or coning error. The algorithm is based on

the Singular Value Decomposition (SVD) method, and makes use of the attitude rates estimated by the SpinUKF. The last fine attitude estimation algorithm, the SSAS misalignment estimator, is also based on the SVD method and estimates the relative misalignment between the SSC and SSAS. These two algorithms are tested for real data from Arase for the first time in this article.

## 3. Magnetometer Calibration

The GAS is located onboard the Arase (Figure 2). Thus its measurements are corrupted with different errors caused by the nearby electronics and magnetic torquers (MTQs). We assume three types of errors are in effect for the GAS: bias, scale factors, and nonorthogonality.

The model for GAS measurements is,

$$\boldsymbol{B}_k = (I_{3\times3} + D)^{-1}\big(A_k\boldsymbol{H}_k + \boldsymbol{b}_k + \boldsymbol{v}_{m,k}\big) \tag{1}$$

where, $\boldsymbol{B}_k$ is the magnetometer measurement vector, $A_k$ is the attitude matrix for the spacecraft, defining the attitude of the spacecraft body frame with respect to inertial frame, $\boldsymbol{H}_k$ is the Earth's magnetic field in the inertial frame, $D$ is a symmetric matrix with 6 parameters that reflect the scaling, nonorthogonality and soft iron errors, $\boldsymbol{b}_k$ is the bias vector and $\boldsymbol{v}_{m,k}$ is the Gaussian zero-mean measurement noise with covariance $\Sigma_k$ as $\boldsymbol{v}_{m,k} \sim \mathcal{N}(0, \Sigma_k)$.

### 3.1. Batch Calibration

Two different algorithms are run on-ground to correct the collected GAS measurements. The first of these is the TWO-STEP algorithm [10]. It estimates a vector formed of 9 parameters: 3 for bias vector, $\boldsymbol{b}_k$, and 6 for the terms of the symmetric $D$ matrix.

Essentially we run the TWO-STEP with a batch of GAS data collected over the 1st perigee pass of the spacecraft (20 December 2016—about 20:30 UTC). The raw measurements, $\boldsymbol{B}_{raw}$, are corrected using the estimated bias vector, $\hat{\boldsymbol{b}}$, and $\hat{D}$ matrix as:

$$\boldsymbol{B}_{cor} = \big(I_{3\times3} + \hat{D}\big)\boldsymbol{B}_{raw} - \hat{\boldsymbol{b}}. \tag{2}$$

The bias vector, $\hat{\boldsymbol{b}}$, and $\hat{D}$ matrix for the GAS measurements over the 1st perigee pass of the spacecraft are estimated as:

$$\hat{\boldsymbol{b}} = \begin{bmatrix} 304 & -382 & 766 \end{bmatrix}^T \text{nT}; \hat{D} = \begin{bmatrix} 0.0037 & 0.0021 & 0.0029 \\ 0.0021 & 0.0055 & 0.0003 \\ 0.0029 & 0.0003 & 0.0038 \end{bmatrix}. \tag{3}$$

The estimated values suggest that especially the nonorthogonality and scaling errors for the GAS measurements is small.

The TWO-STEP algorithm is run regularly to observe possible deviations in the estimated calibration parameters. Thus far the estimated error values at the regular checks are consistent with the estimated values at the 1st perigee pass and no significant change has been observed.

### 3.2. Real-Time Calibration

Although the batch calibration algorithm is sufficient for coarse attitude estimation there are times that, specifically, the magnetometer bias terms vary. One of these cases is when the MTQs are activated about the perigee passes during the nominal mission. Arase is a Sun-oriented spacecraft. The spacecraft is inertially stabilized and its spin-axis makes $5° \sim 15°$ with the Sun direction. To correct $\sim 1°/\text{day}$ deviation of the spin-axis from this intended direction, the MTQs onboard the spacecraft are activated at certain perigee passes. This affects the GAS measurements and imposes additional biases.

Moreover, the current in the electrical circuits for the solar array panels and batteries changes when the spacecraft is in/out of eclipse. An additional bias in the GAS measurements is observed

especially when the spacecraft is again in the sunlight and the batteries start charging. Although, this effect is small compared to the effects of the MTQ activation, it is observed that it may impose additional bias on the GAS measurements as high as 1000 nT.

To overcome these listed issues a real-time magnetometer calibration algorithm is designed. The algorithm, which is capable of sensing the time-variation in the bias terms, is essentially a Kalman Filter (KF) algorithm [11]. The algorithm is an ordinary linear KF but the measurement matrix is state-dependent and the measurement model, which is given as,

$$
\begin{bmatrix} \widetilde{b}_x \\ \widetilde{b}_y \\ \beta \end{bmatrix} = \begin{bmatrix} 1 & 0 & 0 \\ 0 & 1 & 0 \\ 2B_x - b_x & 2B_y - b_y & 2B_z - b_z \end{bmatrix} \begin{bmatrix} b_x \\ b_y \\ b_z \end{bmatrix} + \begin{bmatrix} \underline{v}_{mx} \\ \underline{v}_{my} \\ \eta \end{bmatrix} \tag{4}
$$

is nonlinear. Therefore, the algorithm is termed as the PSLKF.

$$
\widetilde{b}_{x,k} = \frac{1}{N} \sum_{i=k-N+1}^{k} B_{x,i} \tag{5}
$$

$$
\widetilde{b}_{y,k} = \frac{1}{N} \sum_{i=k-N+1}^{k} B_{y,i} \tag{6}
$$

Here are measurements for the spin-planar bias terms that are formed using the spin-planar GAS measurements [11]. The Gaussian white noise terms for these measurements are given as:

$$
\underline{v}_{mx,k} = \frac{1}{N} \sum_{i=k-N+1}^{k} v_{mx,i} \tag{7}
$$

$$
\underline{v}_{my,k} = \frac{1}{N} \sum_{i=k-N+1}^{k} v_{my,i} \tag{8}
$$

The third and last measurement is [10]:

$$
\beta_k = \|\boldsymbol{B}_k\|^2 - \|\boldsymbol{H}_k\|^2 = 2\boldsymbol{B}_k \cdot \boldsymbol{b}_k - \|\boldsymbol{b}_k\|^2 + \eta_k \tag{9}
$$

Here,

$$
\eta_k = 2(\boldsymbol{B}_k - \boldsymbol{b}_k) \cdot \boldsymbol{v}_{m,k} + \|\boldsymbol{v}_{m,k}\|^2. \tag{10}
$$

For the system equation in the PSLKF, the magnetometer bias terms are modeled as constant parameters, $\dot{\boldsymbol{b}}_k = 0$.

Notice that the real-time magnetometer calibration algorithm does not require attitude knowledge. Thus its implementation is straightforward. The fundamental of the algorithm is making use of the spin motion of the spacecraft to derive two quasi-measurements for spin planar bias terms (Equations (5) and (6) [11]).

Figure 4 gives coarse attitude estimation results using the GAS and SSAS measurements. Two subfigures compare the results with and without real-time GAS bias correction. We clearly see the improvement in the spin-axis direction estimates when the GAS measurements are corrected using the bias estimates by the PSLKF algorithm. Note that the gap in the spin-axis estimation results, which is mostly coinciding with the MTQ activation period, is due to the eclipse.

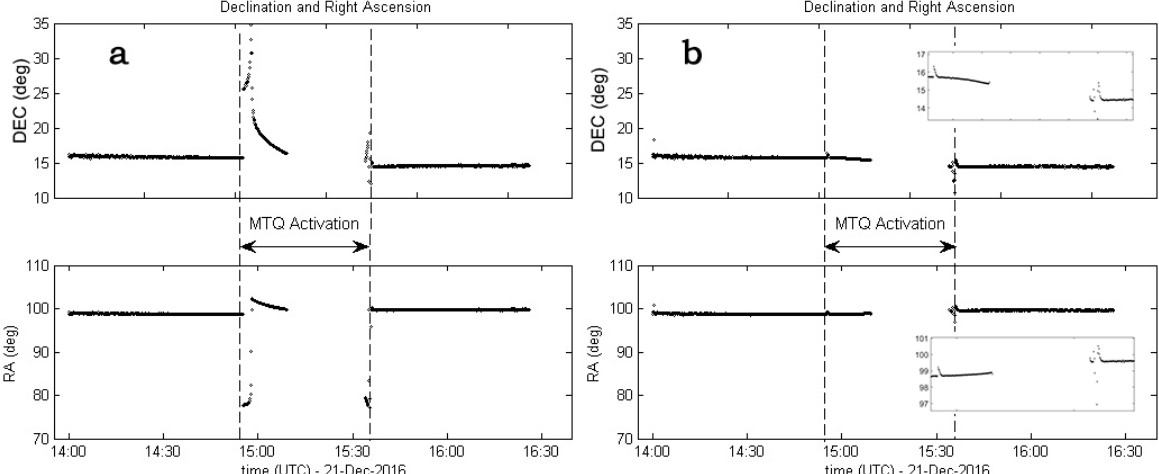

**Figure 4.** Coarse spin-axis estimation over third perigee pass (21 December 2016 about 15:00–15:30 UTC). (**a**) The results without real-time GAS bias correction. (**b**) The results with real-time GAS bias correction. Small plots on figure b shows the zoomed results for DEC and RA estimation starting from 1 min before the MTQ activation and ending at 15:45.

The small bumps in the estimated spin-axis direction at the beginning and end of the MTQ activation in Figure 4b is due to the response of the PSLKF to the suddenly changing bias values. The additional bias vector when the MTQs are fully activated (with maximum torque) is estimated by the PSLKF as,

$$\hat{b} = \begin{bmatrix} -9800 & 400 & -8500 \end{bmatrix}^T \text{nT} \tag{11}$$

and these values are in agreement with the results of MTQ-GAS interference analyses conducted prior to the launch. These analyses were offering that the bias due to the MTQ activation would be:

$$b_{cal} = \begin{bmatrix} -9076 & 495 & -7911 \end{bmatrix}^T \text{nT}. \tag{12}$$

Slight difference in the estimated and calculated values are due to the changes in the current in the electrical circuits for the solar array panels and batteries, which were not accounted for during the analyses.

## 4. Coarse Attitude Estimation

Once the GAS measurements are corrected, the coarse attitude of the spacecraft is determined using these measurements together with the SSAS measurements. The main algorithm that we use for coarse attitude estimation is TRIAD [3,16]. The algorithm is implemented assuming that the SSAS measurements are more accurate compared to the GAS measurements.

Figure 5 gives the estimated spin-axis direction about the 15th perigee pass (26 December 2016 from 6:30 to 9:00 UTC). Perigee pass is right after attitude maneuver that is performed using the RCS. For a period during the attitude maneuver both the SSASs are operational, so there are two different attitude estimations.

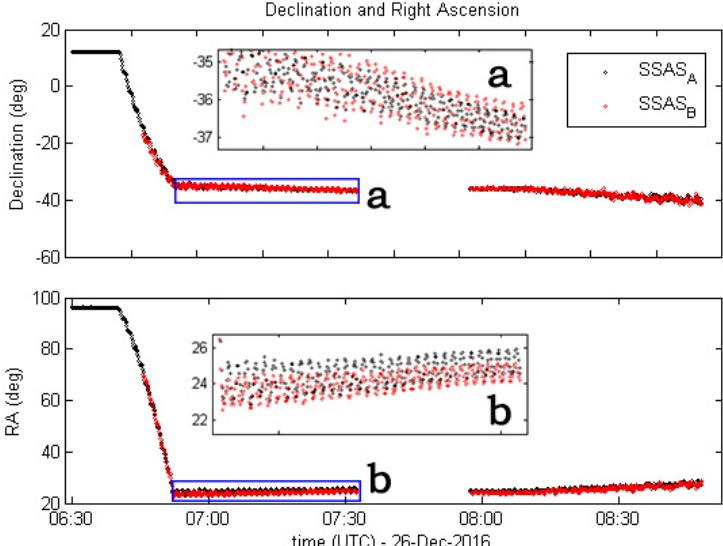

**Figure 5.** Coarse spin-axis estimation about fifteenth perigee pass (26 December 2016—from 6:30 to 9:00 UTC) using GAS and SSAS measurements. Subfigures a and b zoom to the shown parts.

In Figure 5, especially the zoomed sub-figures a and b clearly show the nutation right after the maneuver. Note that on 26 December 2016 only the solar array panels are deployed; the wire antennas and masts are not deployed yet. Moreover, the spin-axis estimations by SSAS-A and SSAS-B do not completely match, and there is shift between the two estimations. This is due to the misalignment of the SSASs and bias in their measurements.

The misalignment/bias for the SSAS-A is estimated and corrected at a later stage when the SSC measurements are available, with multiple star sightings. This will be presented later in the fine attitude estimation section in this paper.

As mentioned Arase has a highly elliptical orbit. The GAS measurements over and around the apogee are usually not usable for attitude determination. Therefore, we prefer using the GAS measurements only over the perigee passes, which is defined as the period when the magnitude of magnetic field is larger than 5000 nT. As the GAS measurements are noisy, we use the reference magnetic field values to decide this interval such that $|H_k| > H_{per}$, where $H_{per} = 5000$nT is the defined threshold for the magnitude of the reference magnetic field.

Despite this fact, throughout the initial mission phase it was possible to estimate the spin-axis attitude of the spacecraft over the apogee passes with an expected accuracy of $1.5°$ $(1\sigma)$ using the GAS measurements. This accuracy is calculated by comparing the estimates with the spin-axis estimation over the perigee pass, which is the most accurate estimate we have in the absence of SSC measurements. Later on for specific apogee passes, the accuracy gradually decreased and became more than $20°$ $(1\sigma)$ due to the solar activity and its effects on the Earth's magnetic field. Investigating this phenomenon is one of the mission goals for Arase.

*Spin Rate Calculation*

The calculation of spacecraft's spin rate is straightforward using the SSAS measurements:

$$\omega_z^{ssas} = \frac{2\pi}{t_{k+1}^{ssas} - t_k^{ssas}}. \tag{13}$$

Here $t_{k+1}^{ssas}$ and $t_k^{ssas}$ are two consecutive Sun crossing times detected by the SSAS. Note that the calculated spin rate is corrupted with zero-mean white noise $\nu_{st}$ that depends on the Sun-sensor timing error.

Figure 6 gives the spacecraft's spin rate for a period of 4 h on 21 January 2017 that is calculated using the SSAS measurements. For comparison the spin rate calculated using the SSC measurements is also given. Note that spin rate calculation using the SSC measurements is possible only after the star identification.

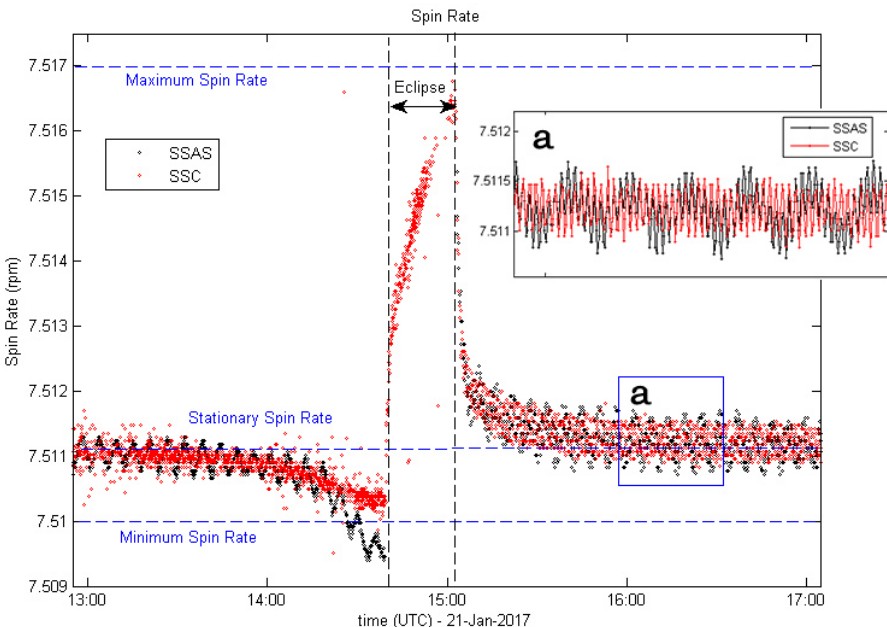

**Figure 6.** Spacecraft's SSAS and SSC calculated spin rate on 21 January 2017 from 13:00 to 17:00 UTC. Subfigure a zooms to the shown part.

Figure 6 shows that the spin rate gradually decreases before the eclipse. This is due to the albedo effect and increase in the length of the wire antennas. Assuming the spin rate of the spacecraft at about the apogee as the stationary spin rate (7.5112 rpm), the increase in the length of the wire antennas, which is 15 m at the nominal condition, is estimated as 8.99 mm. On the other hand, in eclipse the spin rate increases as the spacecraft cools down and wire antennas shrink 50.38 mm in length.

Another interesting feature of Figure 6 is the differences in the SSAS based and SSC based spin rate calculations. Close to the beginning of eclipse SSAS derived spin rate values decreases more than the SSC derived spin rate values. There is always a low frequency oscillation at SSAS based spin rate calculations with a period of 6 m. Such low frequency oscillations are not observable in SSC based spin rate calculations. Reasons of these two phenomena are unclear and still being investigated.

Lastly, as Figure 6 shows there are high frequency oscillations in both SSAS and SSC derived spin rate values. Oscillations have a period about 40 s, due to nutation or in-plane vibrations of the wire antennas. As both motions have similar frequency it is not possible to identify the exact source of the high frequency oscillations in the spin rate values. These oscillations are observed after the eclipse. Therefore, the nutation and/or in-plane wire antenna vibrations are caused by the MTQ activation for attitude maneuver about the perigee pass.

## 5. Star Identification

Star pulses detected by the SSC must be identified (matched with the catalogue stars) before using them as measurements. The overall identification process is composed of star pulse pairing, identification and pulse tagging. For every measurement data set that we receive for a single orbital revolution:

- First the region with the brightest and most star detections is found. This region extends to 10 spin period.

- Measurements for the same stars within this region are paired by comparing their timing, measured elevation and brightness. At the end of the pairing, the averaged values are calculated for each paired measurement. These averaged values are further used for identification.
- Paired (and averaged) star direction vectors for each star are identified and matched with the stars from the catalogue.
- The measurements outside the paired region are searched. If they are matching with one of the identified stars from the paired region they are tagged as the same star.

*5.1. Pairing*

Pairing is the initial step of the star identification process. SSC pulses for 10 consecutive spin periods are paired as a single pulse for the same star with the pairing process. Pulses are paired with their initial detection assuming that pulses for the same star, *l*, will be detected exactly one spin period from each other within a margin of timing error. First detection for star *l* at $t_{l,1}^{ssc}$ is paired with its other detections in 10 consecutive spin periods as,

$$\left| \left( t_{l,k+1}^{ssc} - t_{l,1}^{ssc} \right) - k \times P_{spin} \right| < \xi \tag{14}$$

where $P_{spin}$ is the spin period and *k* is the number of spins after the first detection $k = 1 \ldots 9$. The spin period is estimated using the mean of $\omega_z^{ssas}$. The threshold value, $\xi$, is usually kept small as $\xi = 0.005$s but this caused several difficulties as soon will be discussed.

The pairing for a pulse is accepted successful if the same pulse is detected for more than "*N*" times within 10 spin periods. The default value we choose for "*N*" is 8.

Although, the pairing algorithm relied only on the timing information for SSC pulses, initially, at the later stages of our investigations including the pulse magnitude values ($V_{mag}$) and the elevation angles for the detected stars appeared to be necessary to improve the chance of successful paring and algorithm's robustness. The main reason was the varying spin rate for the spacecraft as was shown in Figure 6. Because of the spin rate variation we needed to increase $\xi$ to have a successful pairing but for specific revolutions this was causing pairing with the wrong stars. $V_{mag}$ and elevation angle thresholds were added as additional measures to prevent faulty pairing even if the timing threshold had to be kept large.

During the pairing process the unit star direction vector measured in the body frame is transformed to the inertial frame using the coarse attitude information. For a set of paired pulses, the mean of this unit star direction vector in the inertial frame is calculated. In the identification process this mean value is used. Also the pulse magnitude values ($V_{mag}$), which indicate the apparent magnitude for the stars, are averaged for the paired star pulses at this phase.

The main advantage of pairing is having misdetections within these 10 spin periods removed and using only valid star detections - in the identification process.

Another difficulty that we had when applying the pairing algorithm was spacecraft's passes through the radiation belts. Within these periods the SSC measurements become highly corrupted and noisy. Thus, in the final form of the algorithm we initiated the pairing outside the radiation belts and even if there is pairing when the spacecraft is within the belts, we did not count this as a successful result and did not use.

*5.2. Identification Process*

Once the pairing is completed, the paired pulses are matched with the catalogue stars. The identification is performed based on the angular distance and orientation comparison. If there are more than one paired star detections, the algorithm first performs the comparison using only the star-to-star distance and orientation. Then the identified stars are validated by checking Sun-to-star angular distances and orientations. On the other hand, if there is only one detected star, the Sun-to-star angular distances and orientations are compared for the identification.

First the vectors $\overset{\rightarrow ij}{V_c}$ in the catalogue (inertial) frame and $\overset{\rightarrow kl}{V_m}$ in the frame for the measurements is calculated using the unit direction vectors for the detected (and paired) stars *i* and *j* and stars *k* and *l* from the catalogue as:

$$\overset{\rightarrow ij}{V_c} = \overset{\rightarrow i}{1_c} - \overset{\rightarrow j}{1_c}; \ \overset{\rightarrow kl}{V_m} = \overset{\rightarrow k}{1_m} - \overset{\rightarrow l}{1_m} \tag{15}$$

The frame for the measurements is the reflected inertial frame. It is formed by transforming the measured star direction vectors in the body frame to the inertial frame using the coarse attitude estimates. $\overset{\rightarrow kl}{V_m}$ vector is formed among each detected star and $\overset{\rightarrow ij}{V_c}$ is formed among each star in the catalogue (Figure 7). Prior to the star identification we limit the candidate stars in catalogue regarding the coarse attitude estimate and SSC's field of view.

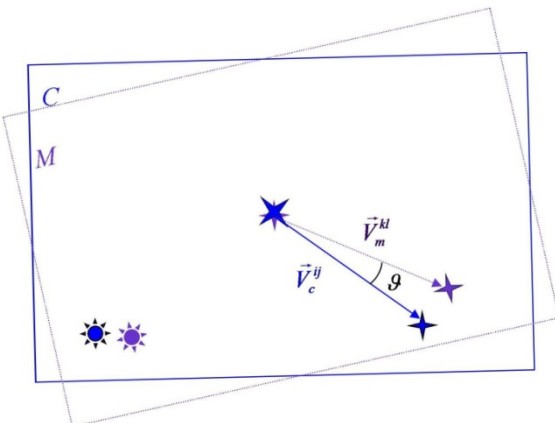

**Figure 7.** Star identification representation in two-dimensional (2D). M is the frame for the measurements and C is the catalogue (inertial) frame.

Star identification is performed by checking the angular distances and cosine similarities (orientation) for each star-to-star couples. If they satisfy certain conditions as,

$$\left| \ \|\overset{\rightarrow ij}{V_c}\| \ - \|\overset{\rightarrow kl}{V_m}\| \ \right| \ \leq \ \varepsilon \tag{16}$$

$$\cos \vartheta = \frac{\overset{\rightarrow ij}{V_c} \cdot \overset{\rightarrow kl}{V_m}}{\|\overset{\rightarrow ij}{V_c}\| \|\overset{\rightarrow kl}{V_m}\|} \ \geq \ 1 - \Delta \tag{17}$$

where $\varepsilon$ and $\Delta$ are the threshold values, than pulses *i* and *j* are identified as stars *k* and *l* from the catalogue, respectively. Note that if $-\cos \vartheta \geq 1 - \Delta$ then pulse *i* is star *l* and pulse *j* is star *k*, instead. Once the stars are identified they need to be verified using the Sun-to-star angular distances and orientations, calculated for each paired star pulse in a similar manner. In this process, the unit direction vectors for one of the stars in Equation (15) is replaced by the Sun direction vector.

The threshold values for the identification algorithm, $\varepsilon$ and $\Delta$, should be selected with respect to various factors, including the accuracy of the coarse attitude estimate. So they may vary for each telemetry dataset. Yet, in general, the threshold for orientation, $\Delta$, is smaller than the threshold for the angular distance, $\varepsilon$. In addition, to make the algorithm more robust, threshold values for the Sun-to-star validation phase are selected separately and are much smaller than those for star-to-star matching phase.

The magnitudes ($V_{mag}$) of the star pulses are not directly used as a part of the identification process due to inaccuracy of these measurements. So far, we observed that the magnitude itself is not reliable for matching a single star with star from catalogue. However, in case there is more than one

star detections the algorithm can verify if $\left\|V_{mag}^{i} - V_{mag}^{j}\right| - \left|V_{mag}^{k} - V_{mag}^{l}\right\| \leq \zeta$, where $\zeta$ is magnitude threshold set as 0.2. We checked the relative magnitudes for two identified stars and compare the relative values from the catalogue and of the measurements.

One major difficulty we had while implementing the identification algorithm was interruption of the planets and moon. They were observed as valid signals for the SSC, which are satisfying the initial pairing check but cannot be identified. In the final version of the algorithm the faulty pulses due to the moon are detected and eliminated. On the other hand, the planet information is added to the star catalogue and their detected pulses are used for attitude determination as well. Five planets are included in the catalogue: Mercury, Venus, Mars, Jupiter and Saturn. Among these planets, Jupiter and Saturn are detected most by the SSC.

When the planets are used as a part of the attitude estimation algorithm the parallax effects should be considered. This is showing the direction change for the planet within one orbital revolution of the spacecraft. If the parallax displacement is large than the planet information is just ignored. Table 1 gives parallax displacements for three of the planets during one spacecraft revolution as an example.

**Table 1.** Parallax displacements for Mercury, Mars and Jupiter during one orbital revolution of Arase.

|  | Mercury | Mars | Jupiter |
| --- | --- | --- | --- |
| Parallax displacement (deg/revo) | 0.99 | 0.41 | 0.09 |

When using the planets detected by the SSC for attitude estimation we need to know the apparent magnitude for the planets. As discussed this information is used as a part of the identification process to minimize the chance of false identification. Apparent magnitude for each planet is calculated by taking the spacecraft-Sun, planet-Sun distances and the triangular layout of planet-spacecraft-Sun into account [17]. For Saturn, this is done by taking the ring effect into consideration. Figure 8 compares the calculated and SSC detected apparent magnitudes for Saturn for several revolutions. Cases before and after the ring effect correction are presented together. Unless this correction is applied, the identification algorithm will fail at identifying the direction measurements associated with Saturn.

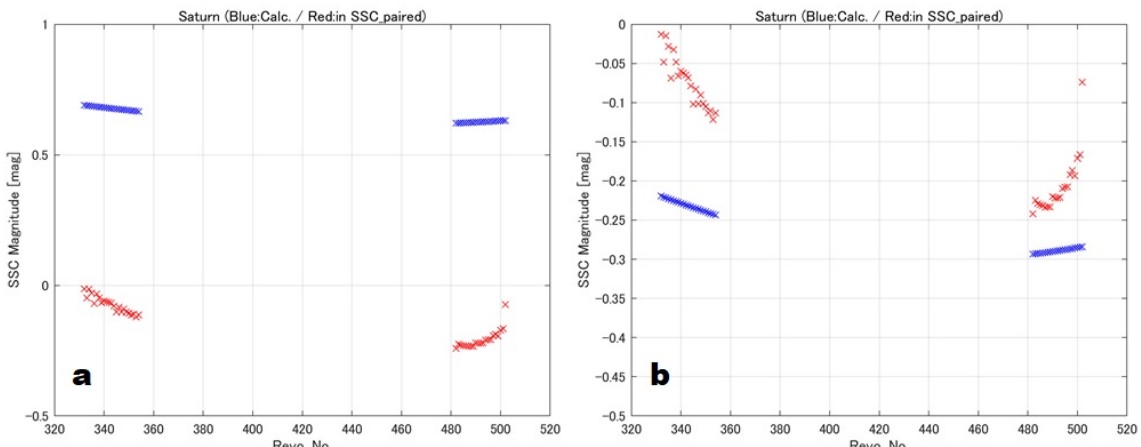

**Figure 8.** SSC detected (red) and calculated (blue) apparent magnitude for Saturn for specific revolutions that it was detected by the SSC. (**a**) Before the ring effect correction for calculated magnitude; (**b**) after the ring correction,

*5.3. Tagging the Star Scanner Measurements*

Once the paired star pulses are identified and matched with the catalogue stars, the star pulses in the rest of the received SSC telemetry data are tagged with the identified star information. The tagging

uses the timing information and checks if the pulse times are the same, within a timing threshold in a spin period. If a pulse at $k$th spin is for star $l$ than is must satisfy the condition.

$$\left| \left( t_{l,k}^{ssc} - k \times P_{spin} \right) - \left( t_{l,i}^{ssc} - i \times P_{spin} \right) \right| < \xi \tag{18}$$

with its previous detections within the last 10 spin periods as $i = (k-10)\ldots(k-1)$ more than "$N$" times to be tagged. Thus the algorithm checks if the same star has been detected in 10 previous spins to tag it. Number of necessary detection to tag a star, "$N$" is selected by the user as $N = 1\ldots 10$.

The time threshold, $\xi$, for tagging might be selected same as the threshold for pairing in Equation (14). However, during the application of tagging algorithm, we had difficulties due to varying spin rate for the spacecraft especially about the perigee passes (see Figure 6). The telemetry data is received as a single pack for one day and necessity for tagging a single star detection for a long period is very likely. This issue was overrun by:

(1) Rather than starting the tagging from the beginning, starting it from the time of paired stars and searching and tagging in both forward and backward directions (Figure 9).

(2) Increasing the threshold linearly by time. $\xi$, which is a constant for pairing, is modified in tagging as $\xi = \xi_0 + \Delta t \xi_t$. Here $\xi_0$ and $\xi_t$ are constant values and $\Delta t$ is the time since last successful tagging for the specific star we are searching for. So, while tagging, we increase the threshold as the search domain moves away from the last tagged measurement. This relaxed time threshold is useful, especially after the periods that SSC cannot detect any star.

(3) Adding other conditions to the tagging based on the magnitudes and elevation angles of the detections. Specifically, for the magnitude check the threshold is adaptively determined as follows:

$$\left| V_{mag,k}^{l} - \sum_{i=1}^{n} V_{mag,i}^{l} \right| \leq 3\sigma_V. \tag{19}$$

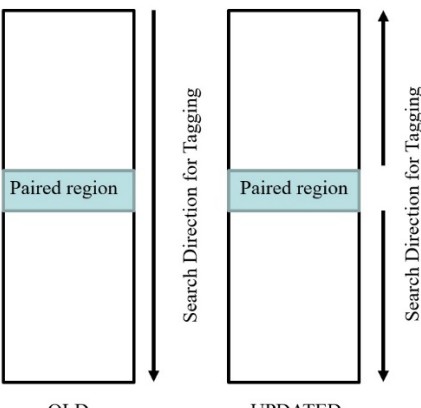

**Figure 9.** Old and updated approaches for tagging the SSC measurements.

Here $V_{mag,k}^{l}$ magnitude of pulse $k$, which is checked to be tagged as star $l$, $V_{mag,i}^{l}$ is the magnitude of $i$th pulse, which is already tagged as star $l$ and whose number is $n$ in total and $\sigma_V$ is the standard deviation of the magnitude for measurements which are already tagged as star $l$.

## 6. Fine Attitude Estimation

Fine attitude estimation block of the proposed filtering-based attitude determination package is used to get the final attitude estimates for Arase based on the SSC and SSAS measurements. These fine attitude estimates are used as the science data is evaluated. The block is composed of the SpinUKF,

an attitude filter that estimates the spin axis direction and attitude rates, the SVD-SAT algorithm for estimating the SAT error and an algorithm for SSAS misalignment/bias correction.

*6.1. SpinUKF*

The SpinUKF is essentially an UKF with the following state components [5]: the spin-axis unit-vector direction terms $\mathbf{1}_i^{spin} = \begin{bmatrix} x & y & z \end{bmatrix}^T$ in the inertial frame, the spin-phase angle $\gamma$, and the body angular rate vector $\boldsymbol{\omega}$ with respect to the inertial frame:

$$\mathbf{X} = \begin{bmatrix} x & y & z & \gamma & \boldsymbol{\omega} \end{bmatrix}^T. \tag{20}$$

The UKF is derived for discrete-time nonlinear equations, so the system model is given by:

$$\mathbf{X}_{k+1} = \boldsymbol{f}(\mathbf{X}_k, k) + \boldsymbol{w}_k \tag{21}$$

$$\mathbf{Y}_k = \boldsymbol{h}(\mathbf{X}_k, k) + \boldsymbol{v}_k \tag{22}$$

Here, $\mathbf{X}_k$ is the state vector and $\mathbf{Y}_k$ is the measurement vector at time $t_k$. Moreover, $\boldsymbol{w}_k$ and $\boldsymbol{v}_k$ are the process and measurement error noises, which are assumed to be Gaussian white noises with covariance matrices $Q(k)$, and $R(k)$, respectively. We estimate the inertial attitude of the spacecraft while the process is propagated by using the discrete-time versions of [5],

$$\dot{x} = \left( \frac{xzs(\gamma) + yc(\gamma)}{r} \right)\omega_x + \left( \frac{xzc(\gamma) - ys(\gamma)}{r} \right)\omega_y \tag{23}$$

$$\dot{y} = \left( \frac{yzs(\gamma) - xc(\gamma)}{r} \right)\omega_x + \left( \frac{yzc(\gamma) + xs(\gamma)}{r} \right)\omega_y \tag{24}$$

$$\dot{z} = -rs(\gamma)\omega_x - rc(\gamma)\omega_y \tag{25}$$

$$\dot{\gamma} = \omega_z - \frac{z}{r}\big(s(\gamma)\omega_y + c(\gamma)\omega_x\big) \tag{26}$$

and the Euler's dynamics equation which is required in the absence of gyros,

$$\dot{\boldsymbol{\omega}} = J^{-1}[\boldsymbol{N} - \boldsymbol{\omega} \times (J\boldsymbol{\omega})]. \tag{27}$$

Here $c(\cdot)$ and $s(\cdot)$ are $\cos(\cdot)$ and $\sin(\cdot)$ functions, respectively, $r = \sqrt{x^2 + y^2}$, $J$ is the inertia matrix of the spacecraft and $N$ is the torque vector, which is sum of the external disturbance torques such as solar radiation pressure and control torques, if there are any. In our application no torque information is given to the filter.

In addition to the SSC and SSAS vector measurements, the SpinUKF uses the SSC and SSAS derived spin rate measurements, which are calculated as in Equation (13). The propagation time for the filter is 0.1 s when there is no available measurement. When a measurement is received, the states are propagated until that time starting from the last estimation and then the update is performed.

Figure 10 shows the spin-axis estimation results by SpinUKF over the 308th perigee pass (20 April 2017 from 11:00 to 20:00 UTC). This perigee pass offers a nice opportunity to test the filter. Because there are more than 2 reliable star detections by the SSC on 20 April 2017. Although, both stars are not in sight all the time. In the figure, together with the SpinUKF estimates, TRIAD results obtained using the SSC and SSAS measurements are also presented.

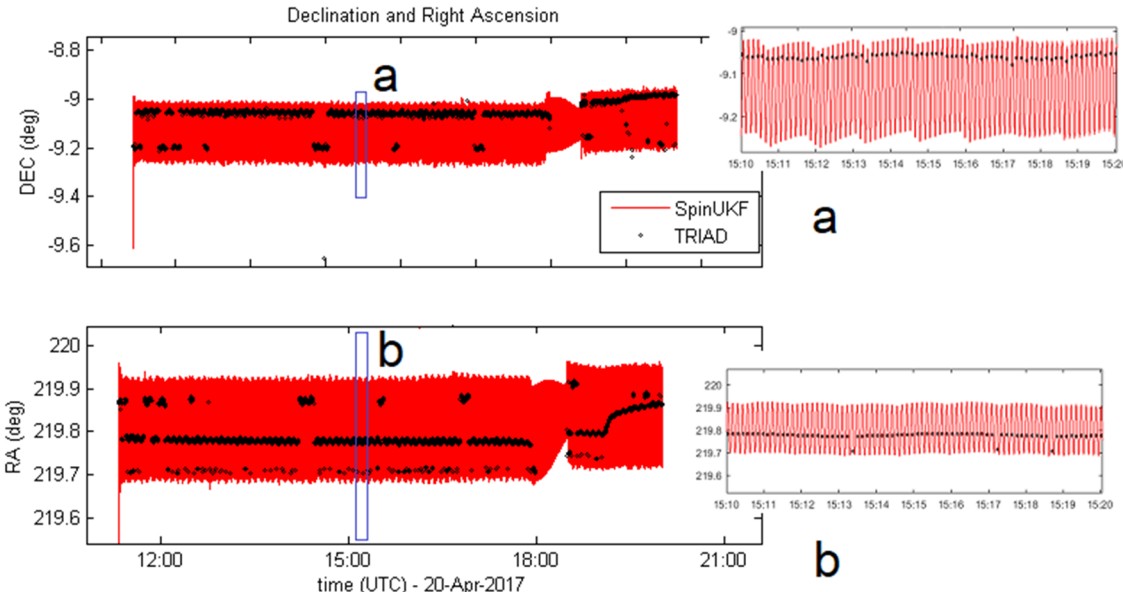

**Figure 10.** Fine spin-axis estimation by the SpinUKF and TRIAD on 20 April 2017 from 11:00 to 20:00 UTC. Both algorithms use SSC and SSAS measurements. Subfigures a and b zoom to the shown parts on main figure.

First of all, Figure 10 shows that the SpinUKF converges to the true attitude quickly. Both the TRIAD and the SpinUKF estimates have oscillations. This is mainly due to the spin-axis tilt. Our previous studies show that spin-axis tilt error causes oscillations and bias in the estimated spin-axis direction [15]. We can observe the oscillations in TRIAD estimates because of the solution at different phases of one spin period.

The TRIAD solutions are single-frame estimates. We usually get single TRIAD solution per spin (period of 8 s). However, the SpinUKF provides continuous spin-axis estimates with a maximum sampling period of 0.1 s. So it can capture the dynamical behavior of body *Z* axis more accurately.

Overall for the SpinUKF the expected accuracy for spin-axis direction estimations is 0.5° or less, depending on the star visibility. This value is calculated with an analysis similar to one presented in [18]. In specific, the estimation algorithm is run for short intervals (1 h each), when no change in the spin-direction is expected. For each interval the starting time is shifted for 10 mins and the process is repeated until the MTQs are activated or SSC and SSAS measurements are interrupted (whichever comes first). Then the mean spin-direction for each interval is calculated. In the end, the variation of attitude estimate, for each of these intervals, are examined and compared with the overall estimate. The residuals that each attitude estimate produces over especially SSC measurements is also checked. This expected accuracy is matching with the mission requirements and far better than the achieved accuracy for a recent similar mission, Van Allen Probes [9]. Yet please consider that there are only magnetometers and Sun sensor onboard the Van Allen Probes for attitude estimation whereas Arase has more accurate SSC.

The SpinUKF has been tested previously with the simulated data [5] and real data [19]. However, this is the first time the filter is applied with non-uniformly sampled SSC measurements. This makes filter tuning rather challenging. Furthermore, real performance of the SSC is still under evaluation. Specifically, we examine the methods to improve the robustness of the filter about the perigee passes. Once these investigations are concluded the filter accuracy may further improve.

Currently we are testing the SpinUKF for different datasets that we received on different dates. This will give us opportunity to understand the drawbacks of filter design and perform filter tuning to improve the estimation accuracy.

### 6.2. Spin-Axis Tilt Estimation

The tilt error is observed as the mismatch between the actual spin axis that corresponds to the maximum principal axis of inertia, i.e., the principal $Z$ axis (or $Z_p$) and the body $Z$ axis (or $Z_b$), see Figure 11, which is the intended spin axis during the design phase. The SAT can be represented as a rotation between the principal and body frames of the spacecraft, and is described by the small Tait-Bryan angles, $\theta_{cx}$ and $\theta_{cy}$ Assuming the rotation angles are small, the rotation matrix that transforms from the principal frame to the body frame is [15]:

$$
C_{bp} = \begin{bmatrix} 1 & 0 & \theta_{cy} \\ 0 & 1 & -\theta_{cx} \\ -\theta_{cy} & \theta_{cx} & 1 \end{bmatrix}.
\tag{28}
$$

The SAT error is expected to be the dominant error in spin-axis direction estimates for Arase. The SpinUKF can estimate the body $Z$ axis orientation in the inertial frame but we need to estimate the SAT error to find spacecraft's real spin axis. This information is crucial for evaluation of the science data.

The objective of the SVD-SAT algorithm is finding the rotation matrix which transforms the angular velocity vector in the body frame (estimated by the attitude filter) to the angular velocity vector in the principal frame. The spin rate around the major principal axis is measured by the SSC using the time log for consecutive star crossings.

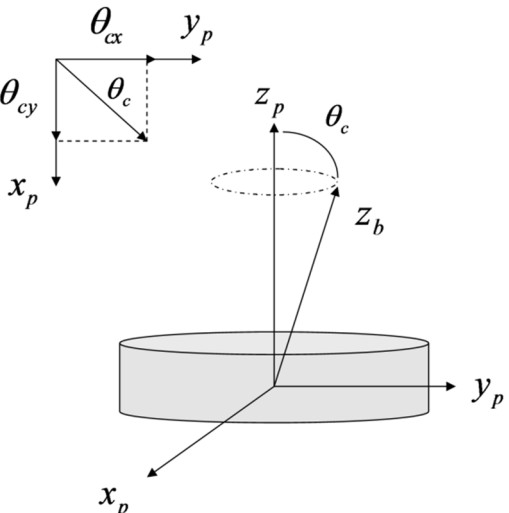

**Figure 11.** Spin-axis tilt (SAT).

The cost function that is to be minimized using the SVD-SAT is:

$$
L(\theta_{cx}, \theta_{cx}) = \frac{1}{2} \sum_l a_l \left| \boldsymbol{\omega}_p^l - \hat{C}_{pb}(\theta_{cx}, \theta_{cy}) \hat{\boldsymbol{\omega}}_b^l \right|^2
\tag{29}
$$

Here the hat operator denotes the estimated quantities, $a_l$ is the sampling weight, $\hat{\boldsymbol{\omega}}_b^l = \begin{bmatrix} \hat{\omega}_{bx}^l & \hat{\omega}_{by}^l & \hat{\omega}_{bz}^l \end{bmatrix}$ is the angular velocity vector in the body frame which is estimated by the SpinUKF filter and $\omega_p^l$ is the angular velocity vector in the principal frame for the $l$th measurement. $\omega_p^l$ is given as,

$$
\omega_p^l = \begin{bmatrix} 0 & 0 & \omega_z^{ssc} \end{bmatrix}^T
\tag{30}
$$

$\omega_z^{ssc}$ is the spin rate of the spacecraft measured by the SSC, in a similar manner with Equation (13).

For SAT estimation the SpinUKF is run using only the SSC measurements for a period when more than one star is detected by the SSC. The main reason is the higher accuracy of the SSC, compared to the SSAS, and the precaution for not being affected by possible SSAS misalignments. Note that the misalignment error here includes the effects for both the alignment errors for sensor's frame with respect to the spacecraft frame and sensor biases. In fact these two errors are indistinguishable for a spinning spacecraft [20]. For the SSC we expect much lower combined sensor bias and misalignment error ($\theta_\beta^{ssc} \cong 0.1°$ for star aspect angle and $\theta_\alpha^{ssc} < 0.045°$ for azimuth at 7.5 rpm spin rate) compared to those for the SSAS ($\theta_\beta^{ssas} \cong 0.5°$ for Sun aspect angle and $\theta_\alpha^{ssas} \cong 0.11°$ for azimuth at 7.5 rpm spin rate).

The cost function in Equation (29) is minimized using the SVD method. The main advantage of the SVD over other algorithms is that it provides numerically robust solutions. The SVD method is well-known [21] so we do not present it here for brevity.

The spin-axis tilt for Arase is estimated as $-0.0986°$ and $0.0476°$ about $x$ and $y$ axes, respectively. These values are mean of the estimation results for different datasets that are received on different dates when the spacecraft is in its final configuration. This estimation involves matching with the observed dynamical behavior of the body $Z$ axis in Figure 10, and also the prior to launch expectations for the SAT error.

Once after we have an estimate for the SAT error we have possibility to estimate the real spin-axis (principal $Z$ axis) of the spacecraft. Figure 12 shows the real spin-axis estimation at again revolution 308. As we clearly see, the real spin-axis of the spacecraft is bounded by the TRIAD estimates for the body $Z$ axis in the inertial frame. This validates the SAT estimation.

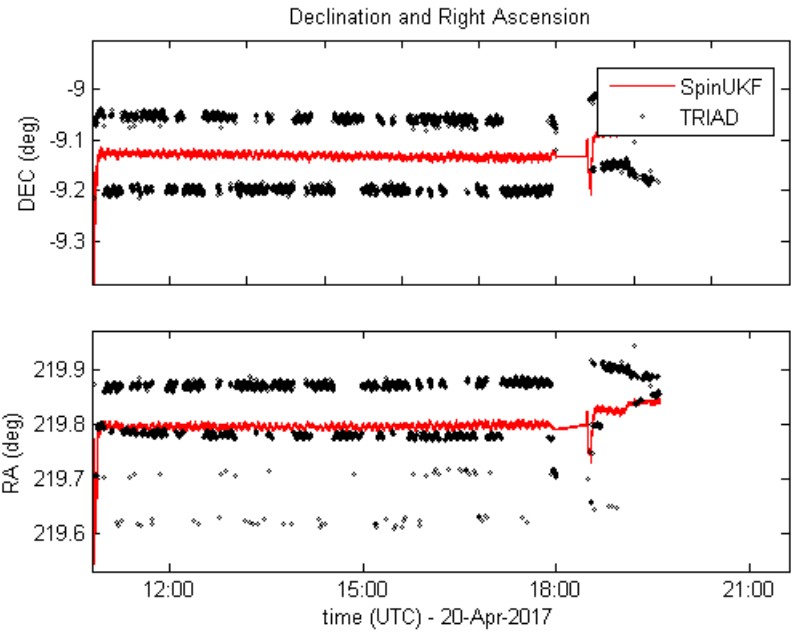

**Figure 12.** Fine spin-axis estimation by the SpinUKF and TRIAD on 20 April 2017 from 11:00 to 20:00 UTC. SpinUKF estimates are for principal $Z$ axis incorporating the SAT correction.

*6.3. Sun Sensor Bias/Misalignment Correction*

Using the SpinUKF attitude estimates, the misalignment of the SSAS frame with respect to the SSC frame can be estimated. The cost function to be minimized is formulated as,

$$L\left(\hat{C}_{ssas}\right) = \frac{1}{2}\sum_l b_l \left|\hat{C}_{ssas}S - \hat{A}_l(x, y, z, \gamma)L\right|^2 \tag{31}$$

where, $\hat{C}_{ssas}$ is the misalignment matrix to be estimated and $b_l$ is the sampling weight. Here we again use the SVD method to minimize the cost function and estimate the $\hat{C}_{ssas}$ misalignment matrix. Note

that SpinUKF is run using only the SSC measurements to estimate the attitude of the spacecraft, $\hat{A}_l(x, y, z, \gamma)$, when we apply the Sun sensor misalignment correction.

Our initial investigations show that the SSAS is misaligned specifically about the sensor's Z axis. Note that this maybe also timing error (bias) as the sensor bias and misalignment are indistinguishable for a spinning spacecraft. The estimated misalignment matrix is:

$$C_{ssas} = \begin{bmatrix} 0.8587 & -0.5034 & -0.0955 \\ 0.4975 & 0.8637 & -0.0793 \\ 0.1225 & 0.0205 & 0.9922 \end{bmatrix}. \tag{32}$$

This is the preliminary result for our SSAS misalignment investigation and yet to be confirmed.

## 7. Conclusions

The details for a filtering-based attitude determination package for JAXA's Arase (ERG) spacecraft and the preliminary attitude estimation results for the spacecraft were presented. Specifically, the star scanner (SSC) performance and the fine attitude estimation algorithms are still under evaluation. Yet, the preliminary results show that the expected accuracy for spacecraft's spin-axis direction estimations is 0.5° or less.

Currently the algorithms are being tested with different datasets that are received on different dates. Moreover, the dynamical phenomena that are introduced in this paper are being analyzed.

**Author Contributions:** H.E.S. and S.-i.S. conducted the research, implemented the algorithms and investigated the results, H.E.S., as well, prepared the manuscript and S.-i.S. supervised with comments and editing; K.A., Y.N., T.T. and I.S., contributed at the level of project administration, support for this specific research and by facilitating the algorithm testing. All authors have read and agreed to the published version of the manuscript.

**Funding:** This research is supported by the Arase (ERG) project of ISAS/JAXA.

**Acknowledgments:** The authors are thankful to the colleagues from NEC Cooperation for their support for receiving and pre-processing the collected sensor data.

**Conflicts of Interest:** The authors declare no conflict of interest.

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
