# Peer review of "Filtering-Based Three-Axis Attitude Determination Package for Spinning Spacecraft: Preliminary Results with Arase"

_aerospace, doi:10.3390/aerospace7070097_

Round 1

Reviewer 1 Report

The article is well written and presents preliminary attitude determination solutions for the Arase spacecraft, taking advantage of its spin-stabilized mode: Attitude-based and attitude-free magnetometer calibration methods, Star identification and fine attitude estimation with SpinUKF. The SpinUKF is a new Unscented Kalman Filter whose state vector is defined via the spin-axis direction vector, the spin-phase angle and the angular velocity. Despite the textual and mathematical clarity of the paper, there are several issues that must be considered and/or addressed:

  1. On numerous occasions the authors state that these results are preliminary and that more work is needed to fully characterize the SpinUKF performance. In particular, lines 466-471 state that the only novelty is the application of the filter with non-uniformly sampled star scanner measurements.  The authors go on to say that it makes filter tuning rather challenging. I recommend for the authors and the editor to reconsider publication of this journal paper considering the fact that results are not final and that the most challenging part is not presented in the paper (or, at least, it was not clear to me).
  2. The abstract fails to give a taste of the advanced attitude determination algorithm (e.g., mention that it is a UKF optimized for spin parameters) and introduce the results (e.g., spin-axis direction accuracy).
  3. The introduction does not include a review of previous work. For example, there are published results (five years ago) of a satellite using an on-board UKF for attitude estimation during high spin rate manoeuvres, including an uncertainty budget. It should also be discussed what is the novelty in magnetometer calibration and in star identification, as compared with the previous research.
  4. Section 3 describes a pseudo-linear Kalman filter (PSLKF) which is used to estimate the time-varying bias in magnetometer measurements, in addition to a simpler TWO-STEP method. However, the results have already been published in [10] and therefore cannot be considered as novel. Please consider whether to include such a section at all.
  5. Figure 4b would benefit from zoom, similar as in Figure 5.
  6. Section 4 presents attitude estimation with TRIAD which is one of the oldest and simplest methods in attitude determination. Please reconsider the inclusion of the section.
  7. Lines 229-235 discuss the attitude estimation ‘error’. It is not clear why such a term is used if the value, as described by authors, is simply a difference between two estimates. By no means such a difference characterises the performance or accuracy. An uncertainty budget should be used for that.
  8. Section 5 presents star identification. Star tracking has been used for decades and it is not clear (from Introduction) how novel is the method. However, the idea of taking the advantage of the spacecraft spin is interesting. Please present the novelty of star identification.
  9. Section 6 presents a new SpinUKF algorithm and preliminary in-orbit results. It is said that the SpinUKF can estimate the spin-axis direction with an accuracy of 0.5 degrees. However, it is not clear where the number comes from. Please include more details and preferably an uncertainty budget in the revision. Of course, the final results would be preferred. 
  10. Figure 10 could benefit from the zoom feature.

Reviewer 2 Report

The paper presents the implementation of an attitude determination methodology and results for a particular spacecraft currently in orbit: Arase. Overall, I found the paper not so easy to ready: the  writing is at times confusing and not very clear. In terms of contribution, there is no theoretical contribution, but it seems like there are practical aspects of the algorithm which are worth sharing. Furthermore, the actual attitude determination results can be potentially novel and interesting to the community.

I have the following specific comments for the authors:

1) The authors need to think of better names/descriptors for the algorithms, instead of 'simple/conventional' and 'advanced/new' and change this throughout the manuscript

2) In Figure 1, please annotate wire antennas

3) Please use the same units (Hz or seconds) to present frequencies or periods (as on p. 3)

4) I was not entirely sure about some of the terminology used in the paper, in particular, 'sun aspect angle', 'spin phase angle'. It would be best if the authors gave precise definitions for these.

5) The statement at the top of p. 4 is not correct: in the absence of torque, the angular momentum of the spacecraft does not change. That does not in general mean that the spin axis direction does not change, unless the initial spin is a principal axis spin.

6) I was confused about the statement about when MTQs were activated (on p. 5, before 3.2 and in the beginning of 3.2). I suggest that the authors give a clear description of the mission progress (initialization and steady-state operation) in the beginning of the paper.

7) I found the real-time calibration in 3.2 to be poorly executed. Why are the authors not using an EKF? This should be addressed or ideally the authors should switch to an EKF.

8) I was not sure about the derivation of equation (7).

9) The authors should explain the gaps that show up in various figures, e.g. Fig. 4 and 5. Are these because magnetometers are not providing measurements at apogee? 

10) I am surprised there was no analysis prior to launch to see the effect of current draw by batteries on the GAS bias. 

11) The explanation of the bumps in Fig 4: are these not because there is a control system which actually makes a correction to the spin axis?

12) Figure captions and legends can be improved to be more clear. For example, in Figure 5, the results are obtained from using both SSAS and GAS? That should be made clear.

13) THe explanation for the error of GAS over the apogee pass given on p. 7 as 'due to the solar activity', are the authors referring to changes in the solar cycle? When they say 'Later on...' what exactly does this mean?

14) The discrepancies noted in Figure 6 close to the beginning of eclipse and the oscillations visible in SSAS but not SSC must be resolved and explained. Could the oscillations be due to thermal effects on wires causing vibrations?

15) I don't see the point of the comments at the end of introductory part for section 4 regarding the T-S algorithm since no results are shown.

16) I was not sure what to make of the results in Figure 8: these need better discussion/explanation and also the relevance of these results to the main focus of the paper.

17) After reading Section 5, I was left unclear about the pairing and tagging processes, in particular, what is the difference between them. Maybe presenting a schematic of when these processes take place in the mission would be helpful.

18) Where do equations 19 come from?

19) A number of equations in the paper are not showing properly, e.g., (24).

20) Overall,  many of the results are still preliminary, as the authors state often themselves. One gets a feeling that the submission is still quite preliminary and the work has not matured. I recommend to the authors to bring to more definitive level several aspects of the research so that more comprehensive results can be presented with confidence and completeness.

Round 2

Reviewer 1 Report

Thanks for improvements! I think Introduction could be improved further by providing a full literature overview and clearly stating the novelty of this paper as compared with previous ones. Also, Conclusions are very short considering the numerous results presented in the paper. However, such 'readability' aspects are up to the authors to decide.